# A nonlinear decomposition analysis of the rural-urban disparities in tobacco use among women in sub-Saharan Africa

Richard Gyan Aboagye[1,2]*, Bright Opoku Ahinkorah[3,4], Irene Esi Donkoh[5], Joshua Okyere[6], Sanni Yaya[7]

1 School of Population Health, University of New South Wales, Sydney, Australia, 2 Department of Family and Community Health, Fred N. Binka School of Public Health, University of Health and Allied Sciences, Hohoe, Ghana, 3 Faculty of Health and Medical Sciences, The University of Adelaide, Adelaide, Australia, 4 REMS Consultancy Services Limited, Sekondi-Takoradi, Western Region, Ghana, 5 Department of Medical Laboratory Science, University of Cape Coast, Cape Coast, Ghana, 6 Department of Allied Health Professions, Sport and Exercise, School of Human and Health Sciences, University of Huddersfield, Huddersfield, England, United Kingdom, 7 The George Institute for Global Health, Imperial College London, London, United Kingdom

* aboagyegyan94@gmail.com

## Abstract

### Background

Tobacco use remains a major public health challenge in sub-Saharan Africa, with significant gendered dimensions. Place of residence is an important determinant, as rural and urban contexts shape exposure, access, and consumption patterns. This study investigates rural–urban disparities in tobacco use among women in sub-Saharan Africa, with a focus on quantifying the relative contributions of socioeconomic factors.

### Methods

We conducted a pooled cross-sectional analysis using nationally representative data from the most recent Demographic and Health Surveys (DHS) of 22 sub-Saharan African countries (2015–2022). The study sample included 350,536 women aged 15–49 years with complete data on tobacco use and relevant covariates. Tobacco use was defined as self-reported current use of cigarettes or other tobacco products. We employed a multivariate decomposition for non-linear response models to quantify the contributions of group differences in characteristics versus differences in how those characteristics affect an outcome. This technique partitions the observed rural–urban gap in tobacco use into two components: (1) endowment effects (compositional differences in characteristics such as education, household wealth, age, marital status, and employment) and (2) coefficient effects (differences in the influence of these

**Data availability statement:** Data for this study were sourced from Demographic and Health Surveys (DHS) and are available at here: http://dhsprogram.com/data/available-datasets.cfm.

**Funding:** The author(s) received no specific funding for this work.

**Competing interests:** The authors declare that they do not have any competing interest.

characteristics on tobacco use between rural and urban women). Models adjusted for sampling weights and survey design effects to ensure representativeness.

## Results

Compositional differences explained 167.48% of the rural–urban disparity in women's tobacco use. Educational attainment and wealth index were the most significant contributors, both showing protective effects. If rural women's education and wealth levels matched those of urban women, tobacco use prevalence would be reduced by 24.99% and 49.84%, respectively. Differences in coefficients accounted for −67.48% of the observed gap, with baseline differences in intercepts (−166.17%) driving most of this effect. These findings highlight both structural disadvantages and variations in behavioural responsiveness across residential settings.

## Conclusion

The study demonstrates that rural–urban disparities in tobacco use among women are primarily shaped by inequalities in education and wealth. Interventions aimed at expanding educational opportunities and addressing poverty in rural communities could substantially reduce tobacco use. Additionally, tailored prevention and cessation strategies targeting women at both the lowest and highest ends of the socioeconomic spectrum are essential to mitigate disparities and advance tobacco control in sub-Saharan Africa.

## Background

In 2010, approximately a third (32.7%) of the world's population aged 15 years or older were active tobacco smokers. However, this decreased to less than a quarter, at 22.3%, in 2020 [1]. The rate is projected to decline to about a fifth (20.4%) of the world's population by 2025, assuming that existing tobacco control initiatives are maintained in all nations [1]. The main objective of the World Health Organization (WHO) Global Action Plan for 2010–2025 is to reduce tobacco use (smoked and smokeless tobacco) by 30% by 2025 compared to 2010 [1]. As such, it has become necessary to monitor the rate of tobacco use. However, this varies across countries, with the best-resourced countries achieving better survey coverage [2].

The rate of tobacco consumption is reducing globally; however, disparities exist in terms of area of residence, sex, and age. WHO stipulates that most tobacco users live in low-and middle-income countries (LMICs), primarily because they lack awareness of the danger associated with it, and sub-Saharan Africa (SSA) is no exception [1,3]. Studies have shown that lower-income earners and less educated individuals are more likely to smoke tobacco than those with higher education levels or higher incomes for both genders in twenty-two sub-Saharan African countries [3,4]. Recent reports also indicate that eight out of these twenty-two countries (Rwanda, Nigeria, Ethiopia, Benin, Liberia, Tanzania, Burundi, and Cameroon) have achieved a 30% reduction in smoking rates [2,3].

Although tobacco use is declining in both sexes, males still have high prevalence globally, but the rate at which it is spreading among women in most sub-Saharan African countries is alarming. This is evident in a study conducted in SSA, where several countries with less than 1% smoking prevalence among female respondents increased from nine in the initial surveys to sixteen in the most current surveys [4,5]. Considering this rate of spread and the retarded developmental situation in SSA, the disparities of tobacco usage among women in rural-urban areas need to be unravelled [2,4]. Existing evidence indicates that tobacco use is more prevalent in women in rural areas [4,5]. This could be due to their lack of education, unawareness of the consequences, occupation, and pride in tobacco usage, exacerbated by their increased free time [6].

While the prevalence of tobacco use calls for attention in both sexes, the consequences in females significantly affect their health, home, and the family due to cognitive impairment [7]. Additionally, tobacco use is associated with several gynaecological issues, including malignancies [8]. There exists an association between tobacco use and breast cancer in women of reproductive age, particularly if the woman starts when she is nulliparous. For women who are seropositive for the Human Papillomavirus (HPV), tobacco use worsens cervical intraepithelial neoplasia and has been linked to cervical squamous cell carcinoma [9,10]. It also increases the chance of early menopause, which in turn raises the risk of cardiovascular disease and osteoporotic fractures [9,11]. In pregnant women, tobacco use increases the chance of many unfavourable pregnancy outcomes such as miscarriage and congenital defects, as well as issues in the offspring such as sudden infant death syndrome and poor lung development in childhood [12,13].

Countries in SSA are working to reduce tobacco use by joining the WHO Framework Convention on Tobacco Control (WHO FCTC), banning tobacco advertising, promotion, and sponsorship, and adding health warnings on tobacco products. However, less than 50% of sub-Saharan African countries have implemented the necessary legislation for all aspects of a tobacco-free policy [14–16]. This could be associated with the fact that SSA remains a desirable location for tobacco industry investment due to its youthful population. Also, the tobacco market in SSA is frequently uncontrolled, cigarettes are inexpensive, and legislation is either ineffective or not properly implemented, even in the urban areas, not to mention the rural areas [5,16]. Given that existing consequences are inevitable in SSA, immediate intervention is necessary to mitigate the existing mortality rate [17].

Previous studies that used the Demographic and Health Survey (DHS) data [3,4] for low- and middle-income countries, including those in SSA, opined on the alarming rate of tobacco use. However, neither of these studies succinctly studies the factors associated with tobacco use per rural-urban strata, which leaves a gap that our study seeks to fill. Also, evidence shows that tobacco use in women of SSA residing in rural-urban areas warrants taking prompt action [4,5]. We decomposed the rural-urban disparity in tobacco use among women in SSA to inform policy and practice aimed at developing and implementing preventive measures to safeguard women's health.

## Methods

### Data source and study design

Data for the study were sourced from the DHS of twenty-two countries in SSA. The study pooled and used data from the individual recode (IR file) in each of these 22 countries. We considered countries for inclusion in the study if their dataset was published from 2015 to 2022 and contained observations on all variables of interest. The DHS provides accurate, nationally representative data on population health, nutrition, family planning, and fertility in over 90 developing countries [18]. A cross-sectional design was used for the DHS. The survey respondents were selected using a stratified two-stage cluster sampling technique, with the detailed sampling methodology highlighted in the literature [18,19]. Pretested structured questionnaires were utilised to gather information from the respondents. All methods were performed in accordance with relevant guidelines and regulations. A weighted sample of 350,536 women aged 15–49 years was included in the study (Table 1). The paper was written with reference to the Strengthening the Reporting of Observational Studies in Epidemiology (STROBE) guidelines [20].

**Table 1. Description of the study sample.**

| Countries | Year of survey | Weighted sample (N) | Weighted % |
|---|---|---|---|
| 1. Angola | 2015−16 | 14,084 | 4.0 |
| 2. Benin | 2018 | 15,601 | 4.5 |
| 3. Burundi | 2016−17 | 16,915 | 4.8 |
| 4. Cameroon | 2018 | 14,470 | 4.1 |
| 5. Ethiopia | 2016 | 15,361 | 4.4 |
| 6. Gabon | 2019−21 | 11,275 | 3.2 |
| 7. Gambia | 2019−20 | 11,622 | 3.3 |
| 8. Guinea | 2018 | 10,651 | 3.0 |
| 9. Kenya | 2022 | 31,497 | 9.0 |
| 10. Liberia | 2019−20 | 7,900 | 2.2 |
| 11. Madagascar | 2021 | 18,482 | 5.3 |
| 12. Mali | 2018 | 10,303 | 2.9 |
| 13. Mauritania | 2019−21 | 15,392 | 4.4 |
| 14. Malawi | 2015−16 | 24,058 | 6.9 |
| 15. Nigeria | 2018 | 40,963 | 11.7 |
| 16. Rwanda | 2019−20 | 14,334 | 4.1 |
| 17. Sierra Leone | 2019 | 15,252 | 4.4 |
| 18. Tanzania | 2015 | 12,992 | 3.7 |
| 19. Uganda | 2016 | 18,127 | 5.2 |
| 20. South Africa | 2016 | 8,104 | 2.3 |
| 21. Zambia | 2018 | 13,402 | 3.8 |
| 22. Zimbabwe | 2015 | 9,751 | 2.8 |
| **All countries** | **2015-2022** | **350,536** | **100.0** |

## Variables

Tobacco use was the outcome variable in the study. During the survey, the women were asked to indicate whether they smoke cigarettes, pipes, or other country-specific tobacco smoking products, or if they do not use any tobacco products at all. We used the definite binary responses 'no' and 'yes' in our study. We recoded the response options into 0 = no (not using any tobacco product at all) and 1 = yes (used at least one tobacco product) and employed these in the final analysis [21–25].

Based on the review of the pertinent literature on factors associated with tobacco use [21–25] and the availability of those variables in the DHS dataset, eleven explanatory variables were selected for this study. The variables consisted of the age of the women, level of education, current working status, marital status, watching television, reading newspapers or magazines, listening to the radio, internet use, sex of the household head, wealth index, and geographic sub-region. We used the existing coding for the age of the women, level of education, current working status, current marital status, internet use, sex of the household head, and wealth index as found in the DHS dataset. In the DHS, frequency of reading newspapers or magazines, listening to the radio, and watching television was categorised as "not at all", "less than once a week", and "at least once a week". In our study, women whose response option was "not at all" were categorised as 'no=0', and the remaining response options were merged and coded to form the 'yes=1' category. The countries included in the study were used to develop the geographic sub-region variable. The sub-region categories were Central, Southern, Eastern, and Western SSA. Place of residence was the equity stratifier used in the analysis.

## Statistical analyses

Data analysis was conducted using Stata software version 17.0. We used spatial maps to show the results of the prevalence of tobacco use among the women in SSA. Subsequently, we used a multivariable binary logistic regression to

examine the factors associated with tobacco use at the pooled level. We further segregated the factors associated with tobacco use by place of residence (rural-urban strata). The regression results were presented in a tabular form using adjusted odds ratios (aOR) with their respective 95% confidence intervals (CI). We weighted all the analyses per the DHS guidelines [18]. Statistical significance was set at $p < 0.05$.

To examine the contribution of each explanatory variable to the overall tobacco use in SSA, we employed a decomposition analysis using 'mvdcmp' command in Stata. Specifically, we employed a multivariate decomposition for non-linear response models to quantify the contributions of group differences in characteristics versus differences in how those characteristics affect an outcome [26]. Decomposition analysis is commonly used to measure the contributions to group differences in the average predictions from multivariate models [26]. According to Powers, Yoshioka, and Yun [26], the nonlinear decomposition analytical method splits the components of a group difference in a statistic, such as means or proportions, into two parts: one for compositional differences between groups-endowment component, or differences in characteristics (E), and another for differences in the effects of characteristics, or differences in due to coefficient (C). We employed this approach to assess the disparities in tobacco use between rural and urban women and to ascertain the relative contributions of each explanatory factor to the variation. It also offers an opportunity to examine the contribution of individual variables or factors to tobacco use.

The endowment component (E) explains how much of the rural-urban disparity in tobacco use is due to differences in measurable factors between rural and urban populations. These factors can include:

- Educational level: Urban residents may have higher levels of education, which is often associated with greater awareness of the health risks of tobacco use. Higher education levels in urban areas may lead to lower tobacco consumption.

- Wealth index: Urban populations may have the richest wealth index, which can affect tobacco use in both positive and negative ways. While a higher wealth index might enable more discretionary spending on products like tobacco, wealthier individuals could also have better access to health information and cessation resources, leading to lower tobacco use.

The coefficient component (C) explains how the effects of these characteristics differ between rural and urban areas. In other words, even if both rural and urban populations share similar characteristics (e.g., income, education), the effect of these characteristics on tobacco use may differ across these environments.

- Educational level effect: In urban areas, higher education may lead to a greater reduction in tobacco use compared to rural areas. For example, while education generally decreases tobacco use, the effect could be more substantial in urban settings where there is increased exposure to anti-tobacco campaigns and access to cessation programmes.

- Wealth index effect: In rural areas, even a slight increase in income may lead to increased tobacco consumption due to the fewer alternative forms of entertainment or relaxation. In contrast, in urban areas, higher income promotes healthier lifestyle choices and reduced tobacco use.

In applying the multivariate non-linear decomposition analysis to rural-urban disparities in tobacco use, a significant portion of the disparity can likely be attributed to E, such as education and wealth index. Rural residents often have lower wealth index and lower levels of education, which collectively contribute to higher tobacco use in rural areas. However, C may be crucial. For example, the positive effects of education and wealth index on reducing tobacco use might be stronger in urban settings due to the infrastructure that supports health awareness and cessation programmes. This suggests that even if rural and urban residents had the same levels of education and wealth, urban residents would probably benefit more from those characteristics due to the supportive environment in which they live.

### Ethical consideration

This study did not require ethical clearance because the DHS dataset is publicly accessible. The ICF Institutional Review Board reviewed and approved the standard DHS survey questions and protocols prior to the DHS's creation to make sure

they adhered to the 45 CFR 46 requirements for the protection of human subjects set forth by the U.S. Department of Health and Human Services. During each survey, the women provided either written or verbal informed consent before inclusion in the DHS. Also, participants' anonymity and confidentiality were adhered to during the survey. All the collected data were securely stored with encryption to avoid any data breaches. Additionally, consent was obtained from the health authorities in each country where the DHS was carried out. All methods were performed in accordance with the relevant guidelines and regulations. Before utilising the dataset in this study, permission was sought from the Monitoring and Evaluation to Assess and Use Results Demographic and Health Surveys (MEASURE DHS). A comprehensive ethical principle guiding the DHS can be accessed at http://goo.gl/ny8T6X.

## Results

### Prevalence of tobacco use in sub-Saharan Africa

Fig 1 shows the prevalence of tobacco use among women in SSA. The pooled results showed that the countries with the highest prevalence of tobacco use among women, as shown in the red-shaded sections of the map, were Benin, Burundi, Madagascar, Mauritania, Sierra Leone, and South Africa. In rural SSA, the prevalence of tobacco use was high among women in Angola, Benin, Burundi, Madagascar, and South Africa. In urban SSA, the highest proportions of tobacco use among women were in Gabon, Madagascar, Sierra Leone, South Africa, and Zambia.

### Factors associated with tobacco use among women in sub-Saharan Africa

The results showed that the likelihood of using tobacco increases with age, with the highest odds among women aged 45–49 [aOR = 7.60; 95%CI: 6.42, 9.00]. In both rural and urban areas, higher education was a protective factor against tobacco use [aOR = 0.25; 95%CI: 0.19, 0.31]. Similarly, having a higher wealth index was associated with lower odds of tobacco use in both rural and urban areas. Regardless of the place of residence, being separated was a significant risk factor for tobacco use [aOR = 1.30; 95%CI: 1.11, 1.53]. While watching television [aOR = 0.87; 95%CI: 0.81, 0.94] and listening to the radio [aOR = 0.88; 95%CI: 0.83, 0.95] were protective factors against tobacco use, other media platforms, including reading newspapers or magazines [aOR = 1.22; 95%CI: 1.11, 1.35] and internet use [aOR = 1.40; 95%CI: 1.26, 1.57] were significant risk factors.

Employed women had higher odds of tobacco use in rural areas [aOR = 1.81; 95%CI: 1.67, 1.97], whereas lower likelihood of tobacco use was reported in urban areas [aOR = 0.81; 95%CI: 0.71, 0.92]. Women in rural female-headed households were less likely to use tobacco (aOR = 0.79; 95%CI: 0.72, 0.87), whereas insignificant association was reported among women residing in female-headed households in urban areas. In both rural and urban areas, women from Eastern, Western, and Southern SSA were more likely to use tobacco compared to their counterparts from Central SSA (Table 2).

### Results from the decomposition analysis

Table 3 presents the results of both overall and detailed decompositions, which investigate rural-urban differences while accounting for existing determinants of tobacco use, such as age groups and women's educational levels.

### Difference due to characteristics (E)

As shown in Table 3, the characteristics of the women (E) accounted for 167.48% of the gap in tobacco use among rural and urban women. This indicates that if only the differences in characteristics between rural and urban were considered, the predicted gap would be greater than the observed one. Level of education and wealth index had significant and negative contributions to tobacco use. If the level of education among rural women was raised to the same level as that of urban women, the use of tobacco would be reduced by 24.99%. If the wealth index of rural women was increased to the same level as that of urban women, tobacco use would be decreased by 49.84%. In particular, the

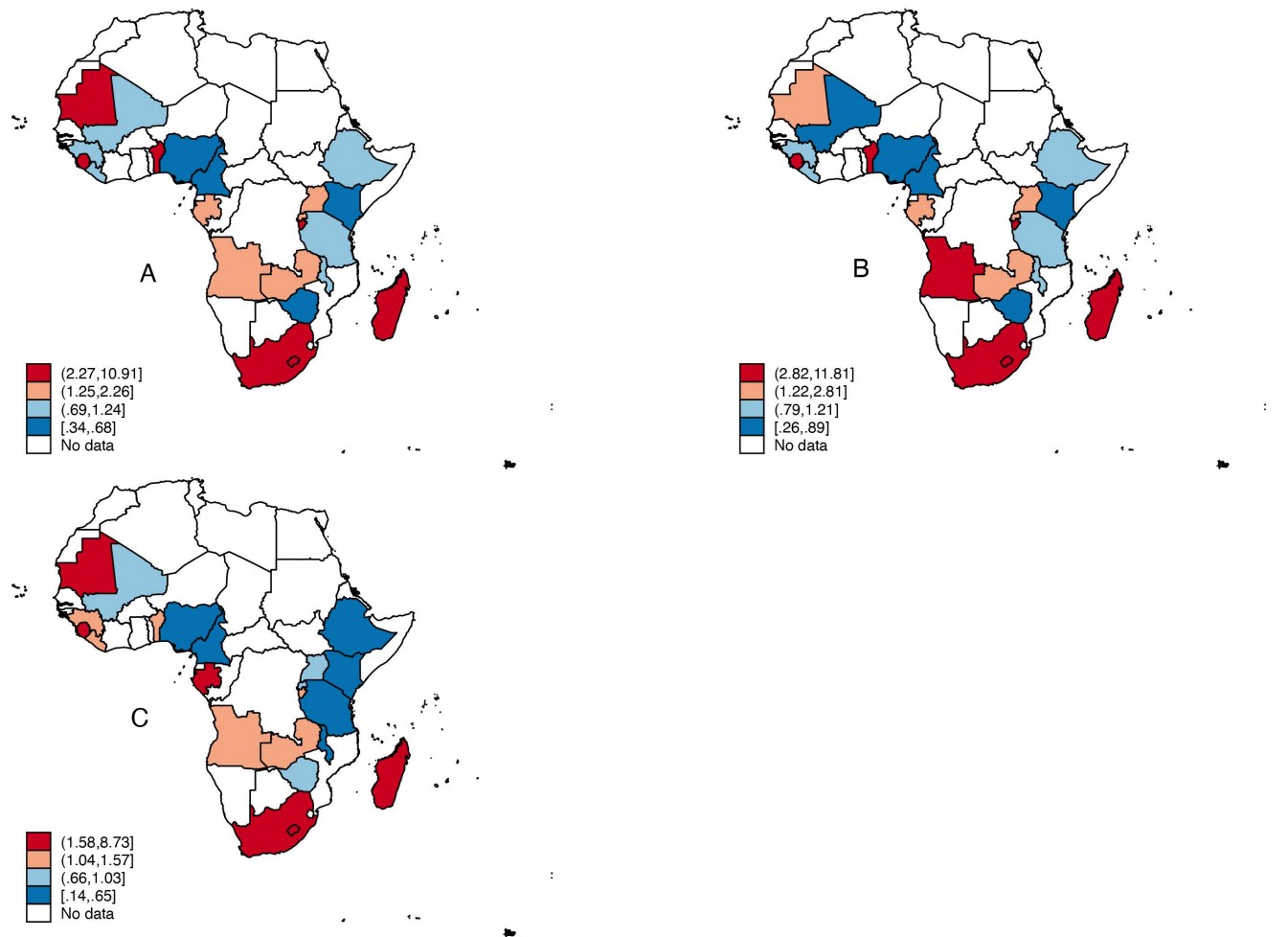

**Fig 1. Propor-on of women who use tobacco in Sub-Saharan Africa (A), Rural SubSaharan Africa (B) and Urban Sub-Saharan Africa (C).**

detailed decomposition (Table 3, differences in characteristics, E) shows that the wealth index is the most significant characteristic accounting for the gap in tobacco use. This is reflected by the largest negative coefficient among the poorest women (−0.00428), indicating that the highest increase in the rural-urban gap would occur if rural women were equal to urban women in the distribution of this characteristic. This means that if rural women were to include the same proportion of the poorest women as urban women, the rural-urban gap in tobacco use observed would be expected to increase by 62.75%. Similar results were obtained among women of the richest wealth index (Coefficient = −0.00340; % = 49.84).

Another variable that significantly contributed to the rural-urban gap was the level of education. With this characteristic, the largest negative coefficient was found among women with no level of education (−0.00246) and those with a higher level of education (−0.00170), which shows that the highest increase in the rural-urban gap is likely to happen if rural women were equal to urban women in the distribution of these characteristics. This means that if rural women included the same proportion of women with no formal education or higher level of education as urban women, the rural-urban gap in tobacco use observed would be expected to increase by 36% and 25%, respectively.

**Table 2. Factors associated with tobacco use among women in sub-Saharan Africa.**

|  | Pooled | Rural | Urban |
|---|---|---|---|
| Variable | aOR [95% CI] | aOR [95% CI] | aOR [95% CI] |
| **Women's age (years)** |  |  |  |
| 15-19 | 1.00 | 1.00 | 1.00 |
| 20-24 | 1.99*** [1.70, 2.34] | 1.83*** [1.51, 2.22] | 1.97*** [1.52, 2.55] |
| 25-29 | 2.81*** [2.38, 3.33] | 2.65*** [2.17, 3.23] | 2.62*** [1.99, 3.46] |
| 30-34 | 3.98*** [3.35, 4.73] | 3.86*** [3.16, 4.71] | 3.49*** [2.56, 4.74] |
| 35-39 | 4.48*** [3.80, 5.30] | 4.62*** [3.80, 5.63] | 3.46*** [2.60, 4.62] |
| 40-44 | 6.17*** [5.21, 7.32] | 6.27*** [5.14, 7.65] | 4.91*** [3.63, 6.63] |
| 45-49 | 7.60*** [6.42, 9.00] | 7.92*** [6.49, 9.67] | 5.57*** [4.14, 7.50] |
| **Women's educational level** |  |  |  |
| No education | 1.00 | 1.00 | 1.00 |
| Primary | 0.62*** [0.57, 0.67] | 0.63*** [0.58, 0.68] | 0.70*** [0.59, 0.84] |
| Secondary | 0.51*** [0.45, 0.56] | 0.46*** [0.39, 0.53] | 0.56*** [0.47, 0.68] |
| Higher | 0.25*** [0.19, 0.31] | 0.23*** [0.14, 0.39] | 0.29*** [0.22, 0.40] |
| **Marital status** |  |  |  |
| Never in union | 1.00 | 1.00 | 1.00 |
| Married | 0.63*** [0.55, 0.71] | 0.79** [0.67, 0.92] | 0.60*** [0.49, 0.74] |
| Cohabiting | 1.04 [0.90, 1.20] | 1.19* [1.00, 1.40] | 1.04 [0.81, 1.34] |
| Widowed | 0.95 [0.80, 1.13] | 1.11 [0.90, 1.36] | 1.12 [0.82, 1.54] |
| Divorced | 0.98 [0.82, 1.18] | 1.16 [0.91, 1.47] | 1.08 [0.81, 1.43] |
| Separated | 1.30** [1.11, 1.53] | 1.55*** [1.28, 1.88] | 1.44** [1.09, 1.89] |
| **Current working status** |  |  |  |
| Not working | 1.00 | 1.00 | 1.00 |
| Working | 1.32*** [1.23, 1.42] | 1.81*** [1.67, 1.97] | 0.81** [0.71, 0.92] |
| **Exposed to watching television** |  |  |  |
| No | 1.00 | 1.00 | 1.00 |
| Yes | 0.87*** [0.81, 0.94] | 0.70*** [0.63, 0.79] | 1.05 [0.91, 1.22] |
| **Exposed to listening to radio** |  |  |  |
| No | 1.00 | 1.00 | 1.00 |
| Yes | 0.88*** [0.83, 0.95] | 0.94 [0.88, 1.01] | 0.87 [0.76, 1.00] |
| **Exposed to reading newspaper or magazine** |  |  |  |
| No | 1.00 | 1.00 | 1.00 |
| Yes | 1.22*** [1.11, 1.35] | 0.86 [0.74, 1.00] | 1.52*** [1.32, 1.76] |
| **Used internet** |  |  |  |
| No | 1.00 | 1.00 | 1.00 |
| Yes | 1.40*** [1.26, 1.57] | 0.98 [0.80, 1.22] | 1.25** [1.09, 1.44] |
| **Sex of household head** |  |  |  |
| Male | 1.00 | 1.00 | 1.00 |
| Female | 0.90* [0.83, 0.98] | 0.79*** [0.72, 0.87] | 1.10 [0.95, 1.28] |
| **Wealth index** |  |  |  |
| Poorest | 1.00 | 1.00 | 1.00 |
| Poorer | 0.75*** [0.69, 0.81] | 0.75*** [0.70, 0.82] | 0.62*** [0.47, 0.82] |
| Middle | 0.66*** [0.61, 0.72] | 0.63*** [0.57, 0.69] | 0.68** [0.53, 0.86] |
| Richer | 0.62*** [0.56, 0.68] | 0.49*** [0.43, 0.56] | 0.62*** [0.49, 0.77] |
| Richest | 0.56*** [0.50, 0.62] | 0.43*** [0.34, 0.53] | 0.53*** [0.42, 0.67] |

*(Continued)*

**Table 2.** (Continued)

| Variable | Pooled | Rural | Urban |
| --- | --- | --- | --- |
| | aOR [95% CI] | aOR [95% CI] | aOR [95% CI] |
| **Geographical sub-regions** | | | |
| Central Africa | 1.00 | 1.00 | 1.00 |
| Eastern Africa | 2.05*** [1.79, 2.35] | 3.06*** [2.59, 3.62] | 1.33* [1.07, 1.67] |
| Western Africa | 1.11 [0.95, 1.29] | 1.21* [1.02, 1.44] | 1.40** [1.12, 1.76] |
| Southern Africa | 2.73*** [2.30, 3.24] | 2.39*** [1.91, 3.00] | 3.30*** [2.62, 4.15] |
| *N* | 350536 | 215508 | 135028 |
| Pseudo *R²* | 0.071 | 0.098 | 0.056 |

Exponentiated coefficients; 95% confidence intervals in brackets; * $p < 0.05$, ** $p < 0.01$, *** $p < 0.001$; aOR = Adjusted Odds Ratio; CI = Confidence Interval.

### Differences due to coefficients (C)

We found that differences in effects account for −67.48% of the observed rural-urban disparity in tobacco use, with differences in intercepts (baseline logits) accounting for the majority of this variation (−166.17%). Current working status, exposure to television, and exposure to newspapers or magazines played a significant role in the total contribution of differences due to the coefficients. For unemployed women, if the proportion of rural and urban women were the same, the rural-urban gap would have reduced by 23% (coefficient = 0.00155; % = −22.77%). For working women, if rural women included the same proportion as urban women, the rural-urban gap would have increased by 35.96%. Similar findings were obtained for exposure to television and newspapers or magazines.

## Discussion

This study decomposed the rural-urban disparities in tobacco use among women in sub-Saharan Africa. The observed prevalence of tobacco use was 2.04%, which aligns with the findings of another study conducted in SSA that found a tobacco use prevalence of 2% [27].

The study revealed that rural-dwelling women used tobacco more than those in urban areas. This finding is corroborated by Dai et al. [28], whose study shows that there is a higher use of tobacco and e-cigarettes in rural areas than in urban areas. Roberts et al. [29] have reported similar rural-urban differences, with tobacco use being more prevalent in rural areas. Women in rural areas may have less access to educational campaigns and information about the health risks associated with tobacco use. Lack of awareness and knowledge about the dangers of smoking can contribute to higher usage rates. Also, the cultural predispositions in rural areas may create an environment that supports tobacco use [30,31]. For instance, the main occupation or source of livelihood for most women residing in rural areas in SSA is agriculture [32]. This type of employment makes tobacco readily available and accessible to women and may influence their use of tobacco. Nevertheless, the findings from this study underscore the need for policymakers in SSA to approach tobacco control programs from a perspective that recognises the factors and the unique situations of rural and urban residents.

Evidence from our decomposition analysis indicates that nearly 167% of the rural-urban differences in tobacco use can be attributed to their characteristics, making it important to recognise the contribution of each characteristic to the identity of the respondents. Irrespective of the place of residence, women of older age were more likely to use tobacco than younger women. This observation aligns with the results of several prior studies [10,27,30] that have explored the association between age and tobacco use, demonstrating a consistent upward trend in tobacco consumption with increasing age. One plausible explanation for this phenomenon is the accumulation of habits and social influences over time. As individuals grow older, they may have been exposed to a longer duration of opportunities to initiate and maintain tobacco

**Table 3. Differences due to characteristics and coefficients in the rural-urban gap in tobacco use.**

| Variable | Difference due to Characteristics (E) | | Difference due to Coefficients (C) | |
|---|---|---|---|---|
| | **Coefficient** | **Percent** | **Coefficient** | **Percent** |
| % Total explained disparity | −0.01142 | 167.48 | 00460 | −67.48 |
| **Women's age (years)** | | | | |
| 15-19 | 0.00004*** | −0.63 | 0.00037 | −5.38 |
| 20-24 | −0.00014*** | 2.01 | 0.00049* | −7.14 |
| 25-29 | −0.00003 | 0.46 | 0.00013 | −1.87 |
| 30-34 | 0.00000* | −0.07 | −0.00007 | 1.03 |
| 35-39 | −0.00004*** | 0.55 | −0.00011 | 1.56 |
| 40-44 | −0.00013*** | 1.84 | −0.00015* | 2.26 |
| 45-49 | −0.00020*** | 3.00 | −0.00019** | 2.72 |
| **Women's educational level** | | | | |
| No education | −0.00246*** | 36.07 | −0.00052 | 7.70 |
| Primary | −0.00081*** | 11.88 | 0.00022 | −3.17 |
| Secondary | −0.00026 | 3.77 | 0.00029 | −4.26 |
| Higher | −0.00170*** | 24.99 | −0.00000 | 0.07 |
| **Marital status** | | | | |
| Never in union | −0.00025 | 3.73 | 0.00003 | −0.44 |
| Married | 0.00113*** | −16.50 | −0.00079* | 11.57 |
| Cohabiting | 0.00000 | −0.01 | −0.00009 | 1.27 |
| Widowed | 0.00000 | −0.02 | −0.00001 | 0.22 |
| Divorced | 0.00003* | −0.43 | 0.00006* | −0.85 |
| Separated | 0.00004*** | −0.57 | 0.00001 | −0.13 |
| **Current working status** | | | | |
| Not working | 0.00019*** | −2.81 | 0.00155*** | −22.77 |
| Working | 0.00019*** | −2.81 | −0.00245*** | 35.96 |
| **Exposed to watching television** | | | | |
| No | −0.00005 | 0.70 | −0.00137*** | 20.13 |
| Yes | −0.00005 | 0.70 | 0.00052*** | −7.68 |
| **Exposed to listening to radio** | | | | |
| No | −0.00013 | 1.88 | 0.00022 | −3.19 |
| Yes | −0.00013 | 1.88 | −0.00023 | 3.41 |
| **Exposed to reading newspaper or magazine** | | | | |
| No | 0.00056*** | −8.15 | −0.00212*** | 31.13 |
| Yes | 0.00056*** | −8.15 | 0.00030 | −4.42 |
| **Used internet** | | | | |
| No | 0.00053** | −7.80 | −0.00046 | 6.69 |
| Yes | 0.00053** | −7.80 | 0.00004 | −0.65 |
| **Sex of household head** | | | | |
| Male | 0.00011* | −1.66 | −0.00086** | 12.59 |
| Female | 0.00011* | −1.66 | 0.00030** | −4.42 |
| **Wealth index** | | | | |
| Poorest | −0.00428*** | 62.75 | 0.00060* | −8.73 |
| Poorer | 0.00019 | −2.85 | −0.00074** | 10.82 |
| Middle | 0.00032** | −4.68 | −0.00044* | 6.50 |
| Richer | −0.00051*** | 7.51 | 0.00015 | −2.13 |

*(Continued)*

 

Table 3. (Continued)

| Variable | Difference due to Characteristics (E) | | Difference due to Coefficients (C) | |
|---|---|---|---|---|
| | Coefficient | Percent | Coefficient | Percent |
| Richest | −0.00340*** | 49.84 | 0.00012 | −1.69 |
| **Geographical sub-regions** | | | | |
| Central Africa | −0.00143*** | 20.94 | −0.00014* | 1.98 |
| Eastern Africa | −0.00012 | 1.83 | −0.00311*** | 45.59 |
| Western Africa | −0.00015* | 2.21 | 0.00147*** | −21.56 |
| Southern Africa | 0.00030*** | −4.46 | 0.00027*** | −4.03 |
| Constant | | | 0.00570*** | −166.17 |

Exponentiated coefficients; 95% confidence intervals in brackets; * $p<0.05$, ** $p<0.01$, *** $p<0.001$

use. This extended exposure to pro-tobacco environments, marketing, and social networks can contribute to the higher likelihood of tobacco use among older women.

A comparison of our findings with other studies [3,4,10] confirms that education is a protective factor for tobacco use among women in SSA. This association remained constant regardless of the place of residence. Our findings align with Özmen [33], whose study revealed a causal inverse relationship between education and tobacco use. Özmen [33] reports that higher levels of education reduce the risk of tobacco use and tobacco-related outcomes by 23%. The protective effect of education against tobacco use can be attributed to several factors, including increased health literacy, enhanced awareness of the health risks associated with tobacco, improved decision-making skills, and greater access to information about the benefits of tobacco cessation. Education equips individuals with the knowledge and tools necessary to make informed choices about their health and well-being, thereby reducing the likelihood of tobacco use. Education can also empower women to challenge social norms and practices that may encourage or normalise tobacco use [34]. The results from the decomposition analysis also highlight that bridging the rural-urban differences in higher education would reduce tobacco use by 24.99%. This provides impetus for the need to invest in female education within SSA.

Another important finding from our study was the significant association between marital status and tobacco use. Our study demonstrated a consistent association between marital status and tobacco use in both rural and urban areas. Specifically, being married emerged as a protective factor against tobacco use, while being separated was identified as a risk factor for tobacco use. These findings contradict the results reported by Sreeramareddy, Pradhan, and Sin [35] but are in accordance with the study conducted by Boua et al. [36], which indicated that tobacco use was less common among married individuals or those living with partners. Perhaps the protective effect of marriage against tobacco use may be explained by various factors. Marriage often represents stability and emotional support, which can reduce stress and the need for tobacco as a coping mechanism. Additionally, the influence of a spouse or partner may discourage tobacco use, as it often entails shared responsibilities, including decisions related to health and lifestyle. On the other hand, separation from a spouse can be associated with increased stress and emotional challenges [37]; this may result in experiencing a sense of social isolation, hence contributing to the use of tobacco as a coping mechanism.

Higher wealth status was also consistently significant in reducing women's odds of using tobacco, regardless of their place of residence. Similar findings have been reported in Sreeramareddy, Harper, and Ernstsen's [4] study. One school of thought is that women in the poorest wealth index may use tobacco as a means of suppressing hunger or appetite [38]. Wealthier individuals often have greater access to education and health resources, which can increase awareness of the harmful effects of tobacco use and provide more opportunities to engage in healthier lifestyles. Additionally, higher income levels may reduce financial stress and the associated coping mechanisms, such as smoking. Wealthier women may also

reside in environments with stricter regulations and social norms against tobacco use. Conversely, those in lower wealth categories might be more susceptible to tobacco use due to limited access to education, healthcare, and smoking cessation resources, as well as higher exposure to stressors. This is further substantiated by the evidence from our decomposition analysis, which indicates that when the proportion of women in the richest wealth index is equalised in both rural and urban areas, there will be a 49.84% reduction in tobacco use among women. Thus, highlighting a need for pro-poor policies and interventions to improve the wealth status of rural-dwelling women.

We found a significant association between employment status and women's use of tobacco – a result that is consistent with extant literature [35]. The results highlight that employment status has varying effects on tobacco use, with important distinctions observed between urban and rural areas. Our research demonstrates that while being employed increases the risk of tobacco use among women overall, a closer examination of the data reveals a contrasting pattern. Being employed is a protective factor among urban women but a risk factor for tobacco use among rural residents. This finding may be attributed to the types of employment available to women in both rural and urban areas. Urban-dwelling women are more likely to be employed in professional, civil, or service occupations. Such occupations have been found to reduce the risk of tobacco use [35]. On the other hand, rural-dwelling women may be heavily involved in agriculture and occupations that predispose them to tobacco use [32]. For example, Sreeramareddy et al. [35] reported that women working in agriculture were more likely to use tobacco.

Women who reported using the internet were more likely to use tobacco; this association was only significant for women in urban areas. This is consistent with some studies [39,40] that have found high use of tobacco among people exposed to the internet. The internet is frequently used for the digital marketing of various forms of tobacco [40]. Given the easy accessibility of urban-dwelling women to the internet, they are likely to encounter pro-tobacco content or advertisements online, which could influence their smoking behaviour. Conversely, the internet may provide access to resources and communities that encourage or normalise tobacco use.

Our research reveals that a one-size-fits-all approach may not be the most effective means of addressing the complex issue of tobacco use across SSA. Instead, our results suggest that tailored interventions are needed, focusing on specific sub-regions and populations. For Southern sub-Saharan African countries, much priority must be on urban-dwelling women. In contrast, Eastern sub-Saharan African countries must place rural-dwelling women at the centre of their tobacco cessation programs.

## Implications for policy and practice

Our findings have revealed that rural-dwelling women are more likely to use tobacco. As such, tobacco control programs should focus on improving awareness and knowledge about the health risks associated with tobacco use in rural areas. Given the widespread availability of internet access in urban areas, there is a need to monitor and regulate digital marketing and online content related to tobacco products. Evidence from the decomposition analysis suggests a need for targeted educational interventions aimed at improving the level of education among rural women, with a focus on increasing awareness about the health risks associated with tobacco use. Additionally, there is a need for pro-poor interventions and policies to improve the wealth status of both rural and urban-dwelling women. Practically, this can be achieved by developing economic initiatives, such as skill development programs and microfinance opportunities, to empower rural women economically, thereby contributing to an increase in their wealth index and, subsequently, a reduction in tobacco use. In Southern SSA, particular attention should be directed towards urban women, while countries in Eastern SSA should prioritise the implementation of tobacco cessation programs centred on rural women.

## Strengths and limitations

Tobacco use was self-reported data. Hence, there is the possibility of recall bias and social desirability bias. The use of only 22 countries does not fully represent the whole of SSA. We are unable to elucidate any causation due to the cross-sectional nature of the data used. Another limitation of this study was that it did not specify the type of tobacco used

by women. Hence, the level of understanding of the nuances of rural-urban differences in tobacco use is missing. Also, the dataset used comprises data from different survey years, which may have contributed to the variations in tobacco use across these years, given that we pooled data from 22 countries and the heterogeneity of tobacco policies. Additionally, potential recall bias or misclassification of tobacco use may have influenced the study's results. Despite these limitations, the data used had a large sample that supports the generalisation of the findings to SSA. The inclusion of a decomposition analysis allowed us to estimate the extent to which each characteristic contributes to the rural-urban differences in tobacco use among women in SSA.

## Conclusion

Our findings contribute to the existing body of research on tobacco use, revealing that there are significant rural-urban differences in tobacco use among women in SSA. The study demonstrates that rural–urban disparities in tobacco use among women are primarily shaped by inequalities in education and wealth. Additionally, being younger in age, having higher educational attainment, higher wealth status, and being married were protective factors, regardless of place of residence. Older age, having no formal education, being in the poorest wealth index, and having been separated were consistent risk factors for tobacco use in both rural and urban areas. Exposure to the internet and Southern SSA were exclusive risk factors in urban areas, whereas being employed and residing in Eastern SSA were exclusive risk factors of tobacco use among rural-dwelling women. Interventions aimed at expanding educational opportunities and addressing poverty in rural communities could substantially reduce tobacco use. Additionally, tailored prevention and cessation strategies targeting women at both the lowest and highest ends of the socioeconomic spectrum are essential to mitigate disparities and advance tobacco control in SSA.

## Acknowledgments

The authors thank the MEASURE DHS project for their support and free access to the original data.

## Author contributions

**Conceptualization:** Richard Gyan Aboagye, Bright Opoku Ahinkorah, Sanni Yaya.

**Data curation:** Richard Gyan Aboagye, Bright Opoku Ahinkorah, Sanni Yaya.

**Formal analysis:** Richard Gyan Aboagye, Bright Opoku Ahinkorah, Sanni Yaya.

**Investigation:** Richard Gyan Aboagye, Bright Opoku Ahinkorah, Irene Esi Donkoh, Joshua Okyere, Sanni Yaya.

**Methodology:** Richard Gyan Aboagye, Bright Opoku Ahinkorah, Irene Esi Donkoh, Joshua Okyere, Sanni Yaya.

**Resources:** Richard Gyan Aboagye, Bright Opoku Ahinkorah, Sanni Yaya.

**Software:** Richard Gyan Aboagye, Bright Opoku Ahinkorah, Sanni Yaya.

**Supervision:** Richard Gyan Aboagye, Bright Opoku Ahinkorah, Irene Esi Donkoh, Sanni Yaya.

**Validation:** Richard Gyan Aboagye, Bright Opoku Ahinkorah, Irene Esi Donkoh, Joshua Okyere, Sanni Yaya.

**Visualization:** Richard Gyan Aboagye, Bright Opoku Ahinkorah, Irene Esi Donkoh, Joshua Okyere, Sanni Yaya.

**Writing – original draft:** Richard Gyan Aboagye, Bright Opoku Ahinkorah, Irene Esi Donkoh, Joshua Okyere, Sanni Yaya.

**Writing – review & editing:** Richard Gyan Aboagye, Bright Opoku Ahinkorah, Irene Esi Donkoh, Joshua Okyere, Sanni Yaya.

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
