## [Decision Letter · Decision Letter 0]

24 Feb 2025

Dear Dr. Abogaye,

Thank you for submitting your manuscript to PLOS ONE. After careful consideration, we feel that it has merit but does not fully meet PLOS ONE’s publication criteria as it currently stands. Therefore, we invite you to submit a revised version of the manuscript that addresses the points raised during the review process.

We look forward to receiving your revised manuscript.

Kind regards,

Mohammad Rifat Haider

Academic Editor

PLOS ONE

2.  We notice that your figure is uploaded with the file type ''\Supporting Information''. Please amend the file type to 'Figure'. Please ensure that each Supporting Information file has a legend listed in the manuscript after the references list.

Additional Editor Comments:

Please address the reviewers' comments and resubmit.

Reviewers' comments:

Reviewer's Responses to Questions

**Comments to the Author**

1. Is the manuscript technically sound, and do the data support the conclusions?

Reviewer #1: Yes

Reviewer #2: Yes

2. Has the statistical analysis been performed appropriately and rigorously?

Reviewer #1: Yes

Reviewer #2: Yes

3. Have the authors made all data underlying the findings in their manuscript fully available?

Reviewer #1: Yes

Reviewer #2: Yes

4. Is the manuscript presented in an intelligible fashion and written in standard English?

Reviewer #1: Yes

Reviewer #2: No

Reviewer #1: The manuscript titled "A nonlinear decomposition analysis of the rural-urban disparities in tobacco use among women in sub-Saharan Africa" addresses an important public health concern by exploring the rural-urban disparities in tobacco use among women in SSA. The study employs a robust multivariate non-linear decomposition analysis and provides meaningful insights into the factors contributing to tobacco use disparities. The findings have significant implications for policymaking and targeted interventions. However, there are some areas that require improvement/corrections to enhance the clarity and consistency of the study's presentation.

Reviewer #2: The study employs a sophisticated analytical method (nonlinear decomposition) to explore an important public health issue. The use of DHS data strengthens the generalizability of findings. Some methodological aspects need further clarification, particularly in explaining decomposition results that exceed 100%. Additionally, more discussion on policy implications would enhance the manuscript’s impact.

**Do you want your identity to be public for this peer review?** For information about this choice, including consent withdrawal, please see our Privacy Policy

Reviewer #1: No

Reviewer #2: **Yes: ** Mohammad Niaz Morshed Khan

---

## [Author Response · Author response to Decision Letter 1]

21 Apr 2025

Response to Reviewers’ Comments

Review of manuscript: A nonlinear decomposition analysis of the rural-urban disparities in tobacco use among women in sub-Saharan Africa

General Comments

The manuscript titled "A nonlinear decomposition analysis of the rural-urban disparities in tobacco use among women in sub-Saharan Africa" addresses an important public health concern by exploring the rural-urban disparities in tobacco use among women in SSA. The study employs a robust multivariate non-linear decomposition analysis and provides meaningful insights into the factors contributing to tobacco use disparities. The findings have significant implications for policymaking and targeted interventions. However, there are some areas that require improvement/corrections to enhance the clarity and consistency of the study's presentation.

Response: Thank you. The authors have addressed all the comments raised during the review.

Introduction

The introduction is well-structured and provides adequate background information on the topic.

Response: Thank you.

Methods

Page 9, Line 237: “Hence, even if rural residents have some level of”- Is it some or same? The authors should verify and correct if this is an error.

Response: The authors have corrected this to read “same”.

Results

a. Page 9, Line 259 and Page 11, Line 267: The authors refer to "Figure 1" in the main text. However, the supporting information labels this figure as "Figure 2". This inconsistency could confuse readers. The authors should carefully review and align the figure numbering in both the main text and supporting materials to ensure consistency and clarity.

Response: The authors have corrected the figure numbering.

b. Page 10, line 274-275: “Similarly, being in a female-headed household and having a higher wealth index was associated with lower odds of tobacco use in both rural and urban areas”- According to Table 2, the aOR for the "sex of household head" in urban areas is 1.10 for female headed household, which indicates that being in a female-headed household is associated with higher odds of tobacco use in urban areas, not lower odds. The authors should re-evaluate this statement and revise it to reflect the results accurately.

Response: The authors have corrected this error.

c. Page 10, line 279: “reading newspapers or magazines [aOR = 1.22; 95%CI: 1.11, 1.25]”- In table 2, it is mentioned as [aOR= 1.22; 95% CI: 1.11, 1.35]. The authors should check this and revise it to reflect the results accurately.

Response: The authors have corrected this error.

d. Page 10, line 285-286: “tobacco use with women in Southern Africa were more likely to use tobacco [aOR = 2.73; 95%CI: 2.30, 3.24] in urban areas” is actually the pooled value for Southern Africa. In table 2, for urban areas it is “[aOR = 3.30; 95%CI: 2.62, 4.15]”. The authors should check this and revise it to reflect the results accurately.

Response: The authors have corrected this error.

e. Page10, line 286-287: “whereas in rural areas, women in Eastern Africa reported the highest likelihood of tobacco use [aOR = 3.30; 95%CI: 2.62, 4.15]” is actually the urban value for Southern Africa. According to Table 2, for rural areas in Eastern Africa it is “[aOR = 3.06; 95%CI: 2.59, 3.62]”. The authors should check this and revise it to reflect the results accurately.

Response: The authors have corrected this error.

Discussion

Page 16, line 373-378: The authors could provide a reference for this statement to support it with evidence.

Response: The sentence in lines 373-378 only reiterates the finding of our study. Hence, there is no need to add a reference.

Page 17, line 414-421: The authors could provide reference for these statements to support with evidence.

Response: The authors have supported the sentence in these lines with references.

Scientific Review of Manuscript PONE-D-24-57996

Title: A Nonlinear Decomposition Analysis of the Rural-Urban Disparities in Tobacco Use Among Women in Sub-Saharan Africa

Reviewer’s Comments

Sl# Comment

1 The title accurately reflects the study’s content, but consider specifying "multivariate nonlinear decomposition" for clarity.

Response: The authors have provided the meaning of the multivariate nonlinear decomposition in the methods section (statistical analysis).

2 The introduction provides a strong rationale for the study, but it could benefit from a clearer articulation of the research gap. What specific aspect of rural-urban disparities has not been addressed in previous research?

Response: The authors clearly provided the research gap in the introduction section.

3 The methodology description is clear, but it would be helpful to justify why a nonlinear decomposition approach was chosen over other decomposition methods.

Response: The authors have justified the use of nonlinear decomposition analysis.

4 The results mention that differences in characteristics accounted for 167.48% of the gap in tobacco use. This percentage exceeds 100%, which might need further clarification for readers unfamiliar with decomposition analysis.

Response: The authors have clarified this in the results section.

5 The conclusion emphasizes the importance of education and wealth index in reducing tobacco use disparities. However, consider briefly discussing potential policy implications in the conclusion section.

Response: The authors discussed the policy implications in a separate subsection and additionally provided a summary of policy and recommendations in the conclusion section.

6 The study design and data source are well described. However, it would be beneficial to discuss any limitations of pooling data from multiple countries in SSA, given potential heterogeneity in tobacco control policies.

Response: The authors have indicated this as a limitation in the study.

7 Statistical analysis is robust, but the rationale for choosing specific covariates in the logistic regression model should be expanded. Were there any omitted variables that could influence tobacco use?

Response: The authors reviewed the literature and selected potential variables that could influence tobacco use. Also, only the variables that were available in the DHS dataset across all the countries included were finally selected for the study. Hence, there may be other variables that could influence tobacco use that were omitted from the study. The authors have acknowledged this as a limitation.

8 The use of spatial maps to depict tobacco prevalence is an excellent approach. However, the manuscript does not provide information on how spatial variability was statistically tested. Were any spatial regression models considered?

Response: Thank you for this comment. The authors only presented the proportion of tobacco use in maps. The authors did not perform any spatial regression analysis.

9 The discussion section is comprehensive, but it would benefit from additional citations comparing findings with studies outside SSA. Are these trends consistent globally?

Response: The authors included citations from studies outside SSA in the discussion section. The trends are similar to findings from other countries outside SSA.

10 The limitations section acknowledges key concerns, such as self-reported data and survey year differences. However, there is no mention of potential recall bias or misclassification of tobacco use, which could affect results.

Response: The authors have acknowledged this as a limitation.

11 The authorship contributions are well stated, but the manuscript does not mention whether the authors had any conflicts of interest regarding the topic.

Response: The authors do not have any competing interests.

12 The ethics statement is clear, but it would be helpful to discuss how data protection and participant confidentiality were maintained, especially given the sensitivity of tobacco use data.

Response: The authors have addressed this comment.

Overall Evaluation:

• Strengths: The study employs a sophisticated analytical method (nonlinear decomposition) to explore an important public health issue. The use of DHS data strengthens the generalizability of findings.

Response: Thank you.

• Areas for Improvement: Some methodological aspects need further clarification, particularly in explaining decomposition results that exceed 100%. Additionally, more discussion on policy implications would enhance the manuscript’s impact.

Response: The authors have addressed these concerns.

---

## [Decision Letter · Decision Letter 1]

18 Jun 2025

Dear Dr. ABOAGYE,

Thank you for submitting your manuscript to PLOS ONE. After careful consideration, we feel that it has merit but does not fully meet PLOS ONE’s publication criteria as it currently stands. Therefore, we invite you to submit a revised version of the manuscript that addresses the points raised during the review process.

We look forward to receiving your revised manuscript.

Kind regards,

Mohammad Rifat Haider

Academic Editor

PLOS ONE

Journal Requirements:

**Additional Editor Comments:**

Thank you for addressing the reviewers' comments/feedback. Please address the comments from the Reviewer 1 on correcting any remaining spelling and grammatical errors prior to final publication. For instance, in the track-changed version, there is still a spelling error on page 15, line 354—“uneployed” should be corrected to “unemployed.”

Reviewers' comments:

Reviewer's Responses to Questions

**Comments to the Author**

Reviewer #1: All comments have been addressed

Reviewer #2: All comments have been addressed

2. Is the manuscript technically sound, and do the data support the conclusions?

Reviewer #1: Yes

Reviewer #2: Yes

3. Has the statistical analysis been performed appropriately and rigorously?

Reviewer #1: Yes

Reviewer #2: Yes

4. Have the authors made all data underlying the findings in their manuscript fully available?

Reviewer #1: Yes

Reviewer #2: Yes

5. Is the manuscript presented in an intelligible fashion and written in standard English?

Reviewer #1: Yes

Reviewer #2: Yes

Reviewer #1: I recommend acceptance of the manuscript for publication, as the authors have adequately and appropriately addressed all of my previous comments. However, I encourage the authors to carefully review the manuscript for any remaining spelling and grammatical errors prior to final publication. For instance, in the track-changed version, there is still a spelling error on page 15, line 354—“uneployed” should be corrected to “unemployed.”

Reviewer #2: Thank you for your effort and addressing the reviewer comments. Hope this paper will add new knowledge in the respective field.

**Do you want your identity to be public for this peer review?** For information about this choice, including consent withdrawal, please see our Privacy Policy

Reviewer #1: No

Reviewer #2: **Yes: ** Mohammad Niaz Morshed Khan

---

## [Author Response · Author response to Decision Letter 2]

26 Jun 2025

RESPONSE TO REVIEWERS’ COMMENTS

Review Comments to the Author

Please use the space provided to explain your answers to the questions above. You may also include additional comments for the author, including concerns about dual publication, research ethics, or publication ethics. (Please upload your review as an attachment if it exceeds 20,000 characters).

Response: Thank you.

Reviewer #1: I recommend acceptance of the manuscript for publication, as the authors have adequately and appropriately addressed all of my previous comments. However, I encourage the authors to carefully review the manuscript for any remaining spelling and grammatical errors prior to final publication. For instance, in the track-changed version, there is still a spelling error on page 15, line 354—“uneployed” should be corrected to “unemployed.”

Response: Thank you. The authors have corrected all spelling and grammatical errors in the manuscript.

Reviewer #2: Thank you for your effort and addressing the reviewer comments. Hope this paper will add new knowledge in the respective field.

Response: Thank you.

---

## [Decision Letter · Decision Letter 2]

20 Aug 2025

A nonlinear decomposition analysis of the rural-urban disparities in tobacco use among women in sub-Saharan Africa

PONE-D-24-57996R2

Dear Dr. Abogaye,

We’re pleased to inform you that your manuscript has been judged scientifically suitable for publication and will be formally accepted for publication once it meets all outstanding technical requirements.

Kind regards,

Mohammad Rifat Haider

Academic Editor

PLOS ONE

Additional Editor Comments (optional):

Thank you for addressing all reviewers' comments.

Reviewers' comments:

Reviewer's Responses to Questions

**Comments to the Author**

Reviewer #1: All comments have been addressed

Reviewer #2: All comments have been addressed

2. Is the manuscript technically sound, and do the data support the conclusions?

Reviewer #1: Yes

Reviewer #2: Yes

3. Has the statistical analysis been performed appropriately and rigorously?

Reviewer #1: Yes

Reviewer #2: Yes

4. Have the authors made all data underlying the findings in their manuscript fully available?

Reviewer #1: Yes

Reviewer #2: Yes

5. Is the manuscript presented in an intelligible fashion and written in standard English?

Reviewer #1: Yes

Reviewer #2: Yes

Reviewer #1: The authors have corrected the identified spelling error as well as reviewed the entire manuscript to ensure spelling and grammatical issues have been addressed.

Reviewer #2: The authors addressed all the comments of the reviewer, therefore, the reviewer has no new comments to add.

**Do you want your identity to be public for this peer review?** For information about this choice, including consent withdrawal, please see our Privacy Policy

Reviewer #1: No

Reviewer #2: No

---

## [Editor Report · Acceptance letter]

PONE-D-24-57996R2

PLOS ONE

Dear Dr. Aboagye,

I'm pleased to inform you that your manuscript has been deemed suitable for publication in PLOS ONE. Congratulations! Your manuscript is now being handed over to our production team.

Kind regards,

on behalf of

Dr. Mohammad Rifat Haider

Academic Editor

PLOS ONE